# Oxytocin signaling in the posterior hypothalamus prevents hyperphagic obesity in mice

Kengo Inada[1]*, Kazoku Tsujimoto[1], Masahide Yoshida[2,3], Katsuhiko Nishimori[2,4], Kazunari Miyamichi[1,5]*†

[1]RIKEN Center for Biosystems Dynamics Research, Kobe, Japan; [2]Laboratory of Molecular Biology, Department of Molecular and Cell Biology, Graduate School of Agricultural Science, Tohoku University, Sendai, Japan; [3]Division of Brain and Neurophysiology, Department of Physiology, Jichi Medical University, Shimotsuke, Japan; [4]Department of Obesity and Inflammation Research, Fukushima Medical University, Fukushima, Japan; [5]CREST, Japan Science and Technology Agency, Kawaguchi, Japan

*For correspondence:
k.inada.repository@gmail.com
(KI);
kazunari.miyamichi@riken.jp
(KM)

†Lead Contact

Competing interest: The authors declare that no competing interests exist.

**Abstract** Decades of studies have revealed molecular and neural circuit bases for body weight homeostasis. Neural hormone oxytocin (Oxt) has received attention in this context because it is produced by neurons in the paraventricular hypothalamic nucleus (PVH), a known output center of hypothalamic regulation of appetite. Oxt has an anorexigenic effect, as shown in human studies, and can mediate satiety signals in rodents. However, the function of Oxt signaling in the physiological regulation of appetite has remained in question, because whole-body knockout (KO) of *Oxt* or *Oxt receptor* (*Oxtr*) has little effect on food intake. We herein show that acute conditional KO (cKO) of *Oxt* selectively in the adult PVH, but not in the supraoptic nucleus, markedly increases body weight and food intake, with an elevated level of plasma triglyceride and leptin. Intraperitoneal administration of Oxt rescues the hyperphagic phenotype of the PVH *Oxt* cKO model. Furthermore, we show that cKO of *Oxtr* selectively in the posterior hypothalamic regions, especially the arcuate hypothalamic nucleus, a primary center for appetite regulations, phenocopies hyperphagic obesity. Collectively, these data reveal that Oxt signaling in the arcuate nucleus suppresses excessive food intake.

## Editor's evaluation

Inada and colleagues report results from a series of studies investigating the role of the neuropeptide oxytocin in regulating food intake and body weight. They used combinations of genetics and behavioral studies to demonstrate that oxytocin deletion results in increased food intake and body weight. They further show that deletion of the oxytocin receptor in the posterior hypothalamus causes a similar increase in food intake and body weight. Collectively, these studies support a role for the oxytocin system as a key regulator of energy balance.

## Introduction

Appetite is one of the strongest desires in animals. The consumption of nutritious foods is a primitive pleasure for animals because it is essential for survival. Yet, excessive food intake leads to obesity and increases the risk of disease. Understanding the neurobiological bases of appetite regulation is therefore an urgent issue, given that the body mass index of humans has increased dramatically over the last 40 years (**NCD Risk Factor Collaboration, 2016**).

Decades of studies in rodents have revealed molecular and neural circuit bases for body weight homeostasis (*Andermann and Lowell, 2017*; *Sternson and Eiselt, 2017*; *Sutton et al., 2016*). Classical studies with mechanical or electrical lesioning, as well as recent molecular or genetic dissections, both support the critical roles of appetite regulation by neurons in the arcuate hypothalamic nucleus (ARH), in particular, those expressing orexigenic agouti-related protein (Agrp) and anorexigenic pre-opiomelanocortin (Pomc) (*Choi et al., 1999*; *Ollmann et al., 1997*). These neurons receive both direct humoral inputs and neural inputs of interoception (*Bai et al., 2019*) to regulate food intake antagonistically at various timescales (*Krashes et al., 2014*; *Sternson and Eiselt, 2017*). The paraventricular hypothalamic nucleus (PVH) is one of the critical output structures of the primary appetite-regulating ARH neurons. Silencing PVH neurons phenocopies the overeating effect observed in the activation of Agrp neurons, whereas activating PVH neurons ameliorates the overeating caused by the acute activation of Agrp neurons (*Atasoy et al., 2012*; *Garfield et al., 2015*). Melanocortin-4 receptor (MC4R)-expressing neurons in the PVH are the key target of ARH Agrp and Pomc neurons. PVH MC4R neurons are activated by α-melanocyte-stimulating hormone provided by Pomc neurons and inhibited by GABAergic Agrp neurons, and are supposed to transmit signals to the downstream target regions in the midbrain and pons (*Garfield et al., 2015*; *Stachniak et al., 2014*; *Sutton et al., 2016*).

Although PVH MC4R neurons have been relatively well documented (*Balthasar et al., 2005*; *Garfield et al., 2015*), other PVH cell types may also mediate output signals to control feeding and energy expenditure (*Sutton et al., 2016*). However, little is known about the organization, cell types, and neurotransmitters by which appetite-regulating signals are conveyed to other brain regions. Neural hormone oxytocin (Oxt), which marks one of the major cell types in the PVH, has received attention in this context (*Leng and Sabatier, 2017*; *Onaka and Takayanagi, 2019*). The anorexigenic effect of Oxt has been shown in humans (*Lawson et al., 2015*; *Thienel et al., 2016*), and genetic variations of *Oxt receptor* (*Oxtr*) have been implicated as a risk factor of overeating (*Çatli et al., 2021*; *Davis et al., 2017*). In rodents, Oxt administration has been shown to suppress increases in food intake and body weight (*Maejima et al., 2018*). Pons-projecting Oxt neurons have been shown to be active following leptin administration (*Blevins et al., 2004*), and knockdown of *Oxtr* in the nucleus of the solitary tract has been reported to alter feeding patterns (*Ong et al., 2017*). In addition, *Oxtr*-expressing neurons in the ARH have been shown to evoke acute appetite suppression signals when opto- or chemo-genetically activated (*Fenselau et al., 2017*). Despite the importance of the Oxt-OxtR system in the food intake and homeostasis of body weight (*Maejima et al., 2018*), knockout (KO) and ablation studies still question such findings (*Sutton et al., 2016*; *Worth and Luckman, 2021*). For example, *Oxt* or *Oxtr* KO mice showed increased body weight at around 4 months of age (termed late-onset obesity), while their food intake was not different from that of wild-type mice (*Camerino, 2009*; *Takayanagi et al., 2008*). Diphtheria toxin-based genetic ablation of *Oxt*-expressing cells in adult mice increased the body weight of male mice with a high-fat diet, but not those with normal chow, and in both cases, food intake was unaffected (*Wu et al., 2012b*). To revisit the function and sites of action of Oxt signaling in the regulation of feeding, acute conditional KO (cKO) mouse models would be useful.

Here, we describe *Oxt* cKO phenotypes related to hyperphagic obesity. Our approach offers the following two advantages over previous studies: (i) the *Oxt* gene can be knocked out in adult mice, avoiding the influence of possible developmental and genetic compensations (*El-Brolosy et al., 2019*); and (ii) the manipulation can be restricted to the brain, or even to a single hypothalamic nucleus, providing a resolution that exceeds previous studies. Owing to these advantages, we show that *Oxt* cKO increases both body weight and food intake. The suppression of overeating and overweighting is predominantly regulated by Oxt neurons in the PVH, leaving Oxt neurons in the supraoptic nucleus (SO) with only a minor role. We further show that *Oxtr*-expressing neurons in the posterior part of the hypothalamus, especially the ARH, mediate the overeating-suppression signals generated by Oxt neurons.

## Results

### cKO of PVH *Oxt* increases body weight and food intake

To examine the necessity of *Oxt* for the regulation of food intake, we prepared recently validated *Oxt^{flox/flox}* mice (*Inada et al., 2022*). In this line, Cre expression deletes floxed exon 1 of the *Oxt*

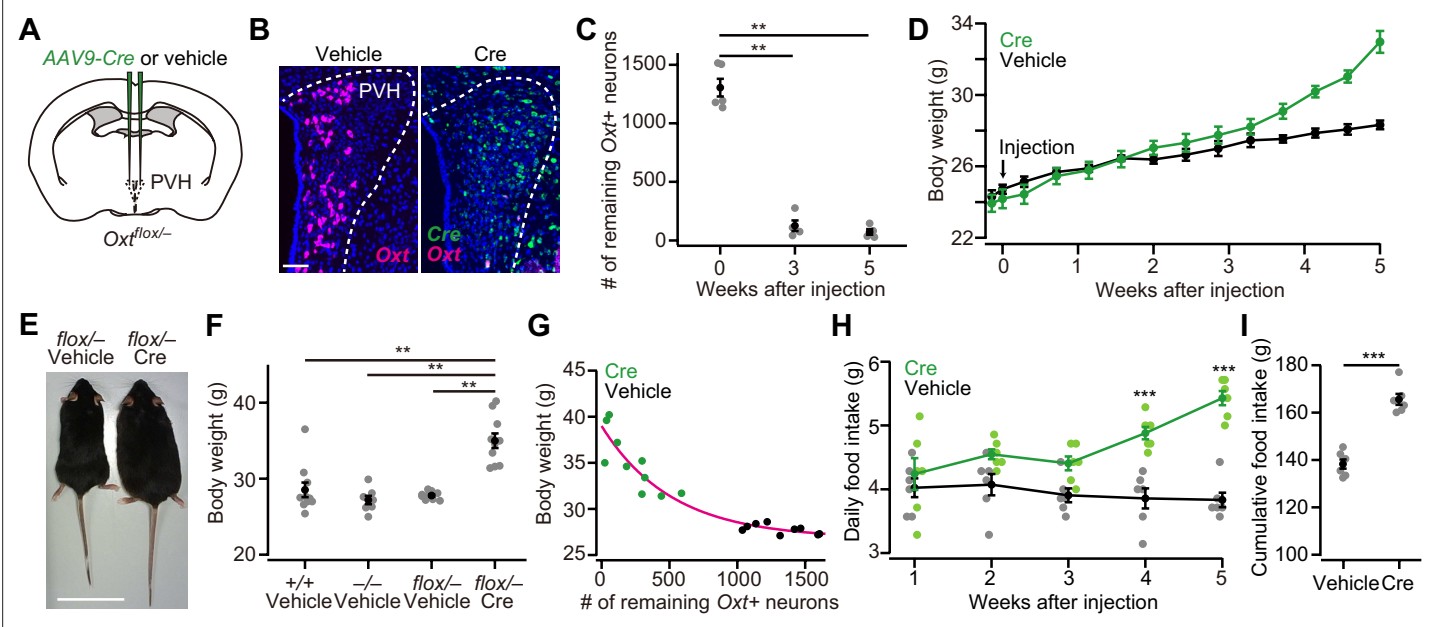

**Figure 1.** *Oxytocin (Oxt)* conditional knockout (cKO) in paraventricular hypothalamic nucleus (PVH) induces an increase in body weight and food intake. (**A**) Schematic of the virus injection. *AAV-Cre* or vehicle was injected into the bilateral PVH of *Oxt^flox/–* male mice. (**B**) Representative coronal sections of the PVH from *Oxt^flox/–* mice received vehicle (left) or *AAV-Cre* (right) injection. Data were obtained at 5 weeks after the injection. Magenta and green represent *Oxt* and *Cre* in situ stainings, respectively. Blue, DAPI. Scale bar, 50 μm. (**C**) The number of remaining *Oxt+* neurons in the PVH of mice that received *AAV-Cre* injection. **p<0.01, one-way ANOVA with post hoc Tukey's HSD. N=5 each. (**D**) Time course of body weight after *AAV-Cre* or vehicle injection. N=6 each. (**E**) Representative photos of *Oxt^flox/–* mice that received either vehicle (left) or *AAV-Cre* injection (right). Five weeks after the injection. Scale bar, 5 cm. (**F**) Body weight of wild-type (*+/+*), *Oxt* KO (*–/–*), and *Oxt* cKO (*flox/–*) mice. The weight was measured at 5 weeks after injection of either vehicle or *AAV-Cre*. Note that this time point corresponds to 13 weeks of age. **p<0.01, one-way ANOVA with post hoc Tukey's HSD. N=10, 7, 9, and 10 for *+/+*, *–/–*, *flox/–* vehicle, and *flox/–* Cre, respectively. (**G**) Relationship between the number of remaining *Oxt+* neurons in the PVH and the body weight of *Oxt^flox/–* mice shown in (**F**). Magenta, exponential fit for the data from both Cre and vehicle. (**H**) Time course of daily food intake, defined as the average food intake in each week after *AAV-Cre* or vehicle injection. ***p<0.001, Student's *t*-test with post hoc Bonferroni correction. N=6 each. (**I**) Cumulative food intake during the 5 weeks after the injection. ***p<0.001, Student's *t*-test. N=6 each. Error bars, standard error of mean (SEM).

The online version of this article includes the following figure supplement(s) for figure 1:

**Figure supplement 1.** *AAV-Cre* injection into the paraventricular hypothalamic nucleus (PVH) of wild-type mice does not induce hyperphagic obesity.

gene, resulting in the loss of transcription of *Oxt* mRNA (**Inada et al., 2022**). To perform the cKO in PVH Oxt neurons, we first crossed *Oxt^flox/flox* and *Oxt* KO (*Oxt^–/–*) mice and obtained *Oxt^flox/–* mice. Then, we injected *AAV-Cre* into the bilateral PVH of 8-week-old *Oxt^flox/–* male mice (**Figure 1A and B**). The number of neurons expressing *Oxt*, visualized by in situ hybridization (ISH), significantly decreased within 3 weeks after the *AAV-Cre* injection (**Figure 1C**). The body weight of *Oxt^flox/–* mice that received *AAV-Cre* injection started to deviate from the controls at around 3 weeks after the injection (**Figure 1D**). At 5 weeks after the injection, we compared the body weight of *Oxt^flox/–* mice that received *AAV-Cre* injection with the wild-type (*Oxt^+/+*), *Oxt^–/–*, and *Oxt^flox/–* mice that received vehicle injection. We found that *AAV-Cre*-injected *Oxt^flox/–* mice were heavier than those in the other groups (**Figure 1E and F**). Importantly, this increase in body weight was considered unlikely to be a reflection of late-onset obesity, as previously reported (**Camerino, 2009**; **Takayanagi et al., 2008**), because we did not find a significant difference between the wild-type and *Oxt^–/–* mice (**Figure 1F**). Next, we analyzed the relationship between the number of remaining *Oxt+* neurons and body weight. We found that mice with a fewer number of remaining *Oxt+* neurons showed a heavier body weight (**Figure 1G**). We also found an increase in food intake: the daily food intake of *Oxt^flox/–* mice that received *AAV-Cre* injection was significantly larger at >4 weeks after the injection (**Figure 1H**), and the total food intake during the 5 weeks after the injection was also larger in the mice that received *AAV-Cre* injection (**Figure 1I**). Of note, these effects are not due to the nonspecific toxicity of *AAV-Cre* injection per se, as *AAV-Cre* injection to wild-type mice did not alter body weight or daily food

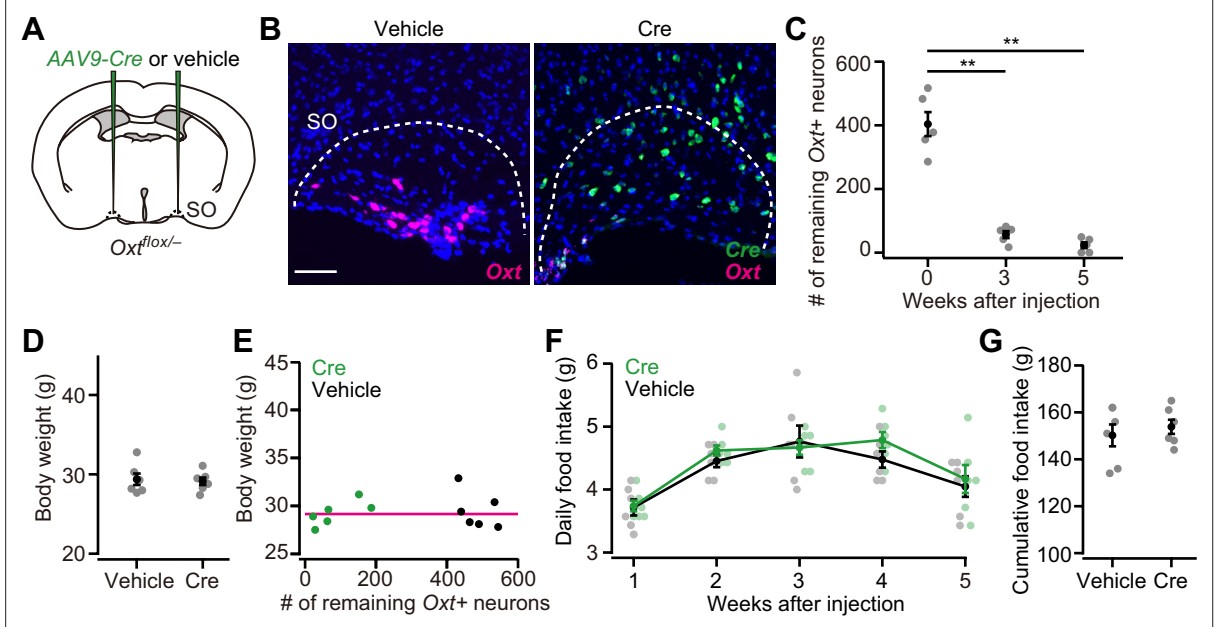

**Figure 2.** *Oxytocin (Oxt)* conditional knockout (cKO) in supraoptic nucleus (SO) has a negligible effect on food intake and body weight. (**A**) Schematic of the virus injection. *AAV-Cre* or vehicle was injected into the bilateral SO of *Oxt^flox/–* male mice. (**B**) Representative coronal sections of left SO from *Oxt^flox/–* mice received vehicle (left) or *AAV-Cre* (right) injection. Five weeks after the injection. Magenta and green represent *Oxt* and *Cre* in situ stainings, respectively. Blue, DAPI. Scale bar, 50 μm. (**C**) The number of remaining *Oxt+* neurons in the SO of mice that received *AAV-Cre* injection. **p<0.01, one-way ANOVA with post hoc Tukey's HSD. N=5 each. (**D**) The body weight of *Oxt^flox/–* mice did not differ between vehicle or *AAV-Cre* (Student's *t*-test). N=6 each. Data were obtained at 5 weeks after the injection. (**E**) Relationship between the number of remaining *Oxt+* neurons in the SO and the body weight of *Oxt^flox/–* mice shown in (**D**). Magenta, exponential fit for the data from both Cre and vehicle. (**F**) The time course of daily food intake was not statistically different (Student's *t*-test with post hoc Bonferroni correction). N=6 each. (**G**) Cumulative food intake during the 5 weeks after the injection. N=6 each. Error bars, SEM.

intake (*Figure 1—figure supplement 1A–D*). These results demonstrate that the cKO of *Oxt* evokes increases in both body weight and food intake.

In addition to the PVH, Oxt neurons are also clustered in the SO (*Zhang et al., 2021*). To examine whether Oxt neurons in the SO also play inhibitory roles on body weight and food intake, we injected *AAV-Cre* into the bilateral SO of *Oxt^flox/–* mice (*Figure 2A and B*). Similar to the PVH, the number of *Oxt+* neurons in the SO significantly decreased at around 3 weeks after the injection (*Figure 2C*). Unlike the PVH, however, neither body weight nor food intake was significantly different compared with controls (*Figure 2D–G*), and no clear relationship was found between the number of the remaining *Oxt+* neurons and body weight (*Figure 2E*). These results suggest that PVH Oxt neurons predominantly regulate food intake and body weight, and that SO Oxt neurons exert little influence.

Because of the minor role of SO Oxt neurons, we focused on the PVH Oxt neurons in the following experiments.

## Weight of viscera and blood constituents

Increased food intake may influence not only body weight, but also the viscera and blood constituents. To examine these points, we collected internal organs and blood samples from non-fasted *Oxt^flox/–* mice that had received either *AAV-Cre* or vehicle injection into the bilateral PVH (*Figure 3A*). While the weight of the stomach was unchanged (*Figure 3B*), a significant increase was observed in the weight of the liver in *Oxt^flox/–* mice with *AAV-Cre* injection, likely because of the accumulation of fat in the liver (*Figure 3C*). We next measured the plasma concentration of glucose, triglyceride, and leptin. No significant differences in glucose levels were found (*Figure 3D*). In turn, the plasma concentrations of triglyceride and leptin were higher in *Oxt^flox/–* mice that had received *AAV-Cre* injection than in those that had received vehicle injection (*Figure 3D*). Of note, a prominent increase in plasma leptin was also reported in the late-onset obesity cases of 6-month-old *Oxt* KO mice (*Camerino, 2009*). Our data regarding *Oxt* cKO showed the plasma leptin phenotype in the earlier stage of

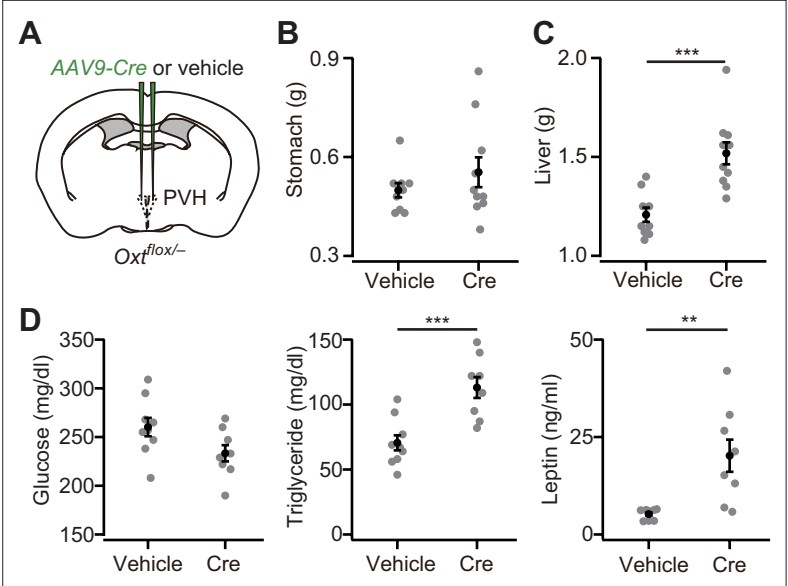

**Figure 3.** Weight of viscera and the blood constituents. (**A**) Schematic of the virus injection. *AAV-Cre* or vehicle was injected into the bilateral paraventricular hypothalamic nucleus (PVH) of *Oxt^flox/–^* male. Data were obtained at 5 weeks after the injection. (**B**) The weight of the stomach was not statistically different (p>0.5, Student's *t*-test. N=9 and 10 for vehicle and Cre, respectively). (**C**) The weight of the liver was significantly heavier in *AAV-Cre*-injected mice (***p<0.001, Student's *t*-test). (**D**) Plasma glucose (left), triglyceride (middle), and leptin (right) measured in the non-fasted *Oxt^flox/–^* mice. **p<0.01, ***p<0.001, Student's *t*-test. N=9 and 8 mice for vehicle and Cre, respectively. Error bars, SEM.

13-week-old mice. These results suggest that the cKO of *Oxt* affects the homeostasis of viscera and blood constituents.

## Oxt supplementation partially rescues *Oxt* cKO

If *Oxt* cKO caused increases in body weight and food intake with higher plasma triglyceride and leptin, such effects might be mitigated by the external administration of Oxt. This hypothesis is also supported by the fact that intraperitoneal (ip) or intracerebroventricular injection of Oxt has been shown to reduce both body weight and food intake (*Maejima et al., 2018*). To examine this hypothesis, first, we injected *AAV-Cre* into the bilateral PVH of *Oxt^flox/–^* mice, and from the next day of the injection, we conducted ip injection of Oxt (100 µL of 500 µM or 1 mM solution) once every 3 days (*Figure 4A*). Ip injection of the vehicle was used as control. At 5 weeks after the injection of *AAV-Cre*, Oxt-treated mice showed significantly reduced body weight, even though the number of remaining *Oxt*+ neurons was not significantly different (*Figure 4B and C*). We found that both daily food intake at 4–5 weeks and total food intake during the 5 weeks after the injection were also significantly reduced (*Figure 4D and E*). The reduction of body weight and food intake by our Oxt treatment paradigm was somehow specific to *Oxt* cKO mice, given that neither reduction of food intake nor body weight was observed in wild-type males (*Figure 4—figure supplement 1A–D*). No significant improvement in the blood samples was found: both plasma triglyceride and leptin tended to be reduced in Oxt-treated mice, but did not reach the level of statistical significance (*Figure 4F*). These results suggest that external administration of Oxt can rescue at least the hyperphagic obesity phenotype of *Oxt* cKO.

In addition to the daily food intake that we have examined so far, previous studies showed that mice ate less within several hours after receiving ip injection of Oxt (*Arletti et al., 1989*; *Maejima et al., 2011*). To examine whether *Oxt^flox/–^* mice that receive *AAV-Cre* injection similarly show reduced hourly food intake, we measured food intake after 6 hr of fasting (*Figure 4—figure supplement 1E and F*). After fasting, the mice received an ip injection of Oxt, and food was provided again (*Figure 4—figure supplement 1F*). Cumulative food intake was measured at 1, 3, and 5 hr after the placement of food (*Figure 4—figure supplement 1F*). Although the number of remaining *Oxt*+ neurons was comparable (*Figure 4—figure supplement 1G*), Oxt-injected mice ate less (*Figure 4—figure supplement*

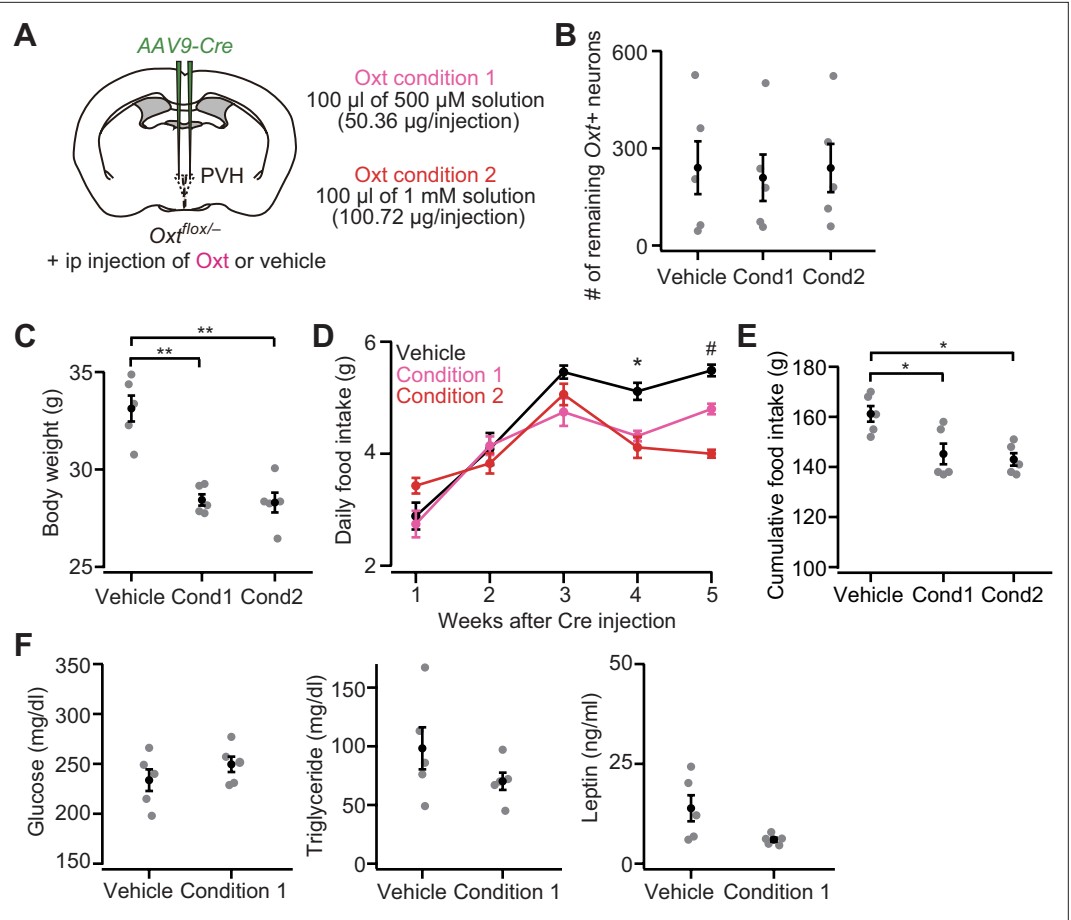

**Figure 4.** Intraperitoneal (ip) injection of oxytocin (Oxt) partially rescues paraventricular hypothalamic nucleus (PVH) *Oxt* conditional knockout (cKO) phenotypes. (**A**) Schematic of the experiments. *AAV-Cre* was injected into the bilateral PVH of *Oxt$^{flox/-}$* male mice. Data were obtained 5 weeks after the virus injection. Once every 3 days, the mice received ip injection of vehicle, 50.36 μg of Oxt (condition 1) or 100.72 μg of Oxt (condition 2) (see Materials and methods). (**B**) The number of remaining *Oxt+* neurons was not statistically different ($p > 0.9$, one-way ANOVA). Cond, condition. N=5 each (same mice across panels **B–E**). (**C**) Ip injection of Oxt significantly decreased body weight. **$p < 0.01$, one-way ANOVA with post hoc Tukey's HSD. Cond, condition. (**D**) Time course of daily food intake. Asterisks (*) denote significant differences for vehicle versus condition 1 and vehicle versus condition 2 ($p < 0.05$, Tukey's HSD), and hashes (#) denote significant differences for vehicle versus condition 1, vehicle versus condition 2, and condition 1 versus condition 2 ($p < 0.05$, Tukey's HSD). (**E**) Cumulative food intake during the 5 weeks after the virus injection was decreased in the mice that received ip injection of Oxt (*$p < 0.05$, one-way ANOVA with post hoc Tukey's HSD). Cond, condition. (**F**) Plasma glucose (left), triglyceride (middle), and leptin (right) measured in the non-fasted *Oxt$^{flox/-}$* mice. Decreases in triglyceride and leptin on average were found in Oxt-treated mice but did not reach the level of statistical significance ($p = 0.314$ and $0.065$ for triglyceride and leptin, respectively, Student's *t*-test). N=5 each. Error bars, SEM.

The online version of this article includes the following figure supplement(s) for figure 4:

**Figure supplement 1.** Intraperitoneal (Ip) injection of oxytocin (Oxt) does not affect either body weight or food intake in wild-type mice, but reduces the hourly food intake of mice with *Oxt* conditional knockout (cKO) in the paraventricular hypothalamic nucleus (PVH).

---

*1H*). Taken together, ip injection of Oxt appears to reduce food intake on the scale of hours to days, thereby preventing the hyperphagic obesity induced by *Oxt* cKO in the PVH.

## *Oxtr*-expressing cells in the ARH mediate appetite suppression

Having established the importance of Oxt to suppress hyperphagic obesity, we examined the site of action of Oxt signaling that mediates appetite suppression. To this end, we prepared *Oxtr$^{flox/flox}$* mice,

in which the *Oxtr* gene can be knocked out under Cre expression (*Takayanagi et al., 2005*). Given that PVH Oxt neurons send dense projection to the hypothalamic nuclei (*Yao et al., 2017*; *Zhang et al., 2021*), and that OxtR expression is also found in the hypothalamus (*Fenselau et al., 2017*; *Mitre et al., 2016*; *Newmaster et al., 2020*), we suspected that a fraction of appetite suppression signals is mediated by the other nuclei of the hypothalamus. To test this possibility, we injected *AAV-Cre* (serotype 9) into the bilateral 'anterior hypothalamus', mainly aiming at nuclei such as the anteroventral periventricular nucleus, medial preoptic nucleus medial part (MPNm), MPN lateral part, and medial preoptic area (*Figure 5A*; see Materials and methods), and the 'posterior hypothalamus', containing nuclei such as the dorsomedial nucleus of the hypothalamus, ventromedial hypothalamic nucleus (VMH), lateral hypothalamic area (LHA), and ARH (*Figure 5B*). AAV-mediated *Cre* expression roughly covered these nuclei (*Figure 5C and D*). To examine if Cre expression reduced *Oxtr* expression, we visualized *Oxtr* mRNA using the RNAscope assay (see the Materials and methods) (*Sato et al., 2020*; *Wang et al., 2012*). As *Oxtr* expression was observed as a dot-like structure (*Figure 5E and F*), we counted the number of such RNAscope dots in each DAPI+ cell. In a negative control experiment utilizing *Oxtr* KO mice, we often detected one or two RNAscope dots in the DAPI+ cells (*Figure 5—figure supplement 1A and B*). Therefore, we regarded a cell with three or more dots as an *Oxtr*-expressing cell (*Oxtr+*; *Figure 5G and H*, and *Figure 5—figure supplement 2A–C*). We found that *AAV-Cre* injection successfully reduced the number of *Oxtr+* cells in most of the targeted nuclei (*Figure 5G and H*). *Oxtr* cKO in the posterior but not anterior hypothalamus significantly increased body weight (*Figure 5I and J*). Similarly, a significant increase in food intake was observed in the mice that had received *AAV-Cre* injection into the posterior hypothalamus (*Figure 5K and L*).

We next aimed to pinpoint a specific nucleus in the posterior hypothalamus that could suppress hyperphagic obesity. To this end, we injected *AAV-Cre* (serotype 2) into the ARH or LHA (*Figure 6A and B*). This serotype of AAV-driven *Cre* expression was spatially localized (*Figure 6C and D*) compared with the AAV serotype 9 used in *Figure 5*. AAV-driven *Cre* expression reduced *Oxtr* expression in *Oxtr^flox/flox* mice (*Figure 6E and F*). Body weight, daily food intake, and cumulative food intake were significantly greater in the mice that received *AAV-Cre* injection into the ARH, whereas no significant difference was found in the mice that expressed *Cre* in the LHA (*Figure 6G-L*).

Taken together, these results indicate that a fraction of the appetite suppression signals from Oxt neurons is mediated by *Oxtr*-expressing cells in the posterior hypothalamus, especially those in the ARH.

## Discussion

### *Oxt* cKO increased body weight and food intake

In this study, we performed cKO of the *Oxt* gene by injecting *AAV-Cre*, which enabled region-specific KO of *Oxt*. By this advantage, we showed that Oxt produced by PVH Oxt neurons contributes to the regulation of body weight and food intake, whereas that by SO Oxt neurons does not (*Figures 1 and 2*). These data extend the previous results that mechanical disruption of PVH in rats increased both body weight and food intake (*Shor-Posner et al., 1985*; *Sims and Lorden, 1986*). In contrast to the *Oxt* cKO phenotype (*Figure 1H and I*), whole-body *Oxt* KO mice showed a normal amount of food intake (*Camerino, 2009*), suggesting compensatory mechanisms. For example, when a certain gene is knocked out, expression of the related gene(s) is enhanced to compensate for some of the KO phenotypes functionally (*El-Brolosy et al., 2019*; *Ma et al., 2019*). Transcriptomic analysis between *Oxt* KO and wild type may reveal a more complete picture of gene expression that can explain the compensational mechanisms in *Oxt* KO mice.

The phenotypic discrepancy between our data and diphtheria toxin-induced ablation of Oxt cells (*Wu et al., 2012b*) might be due to the loss of Oxt cells outside the PVH (maybe even outside the brain; *Paiva et al., 2021*) that somehow elicited appetite suppression, and therefore counterbalanced the overeating phenotype caused by the loss of *Oxt* in the PVH. Alternatively, neuropeptides or neurotransmitters other than Oxt expressed in the PVH Oxt neurons might have conveyed appetite-stimulating signals, which remained intact in our *Oxt*-selective cKO model, but were disrupted in the cell-based ablation, resulting in only the overeating phenotype to appear in our case. Regardless of the scenario, our data establish the necessity of Oxt in the PVH to suppress overeating and suggest the presence of a hormone-based output pathway of PVH appetite regulation signals, in addition to the well-established neural pathways mediated by MC4R neurons (*Garfield et al., 2015*; *Stachniak et al., 2014*; *Sutton et al., 2016*).

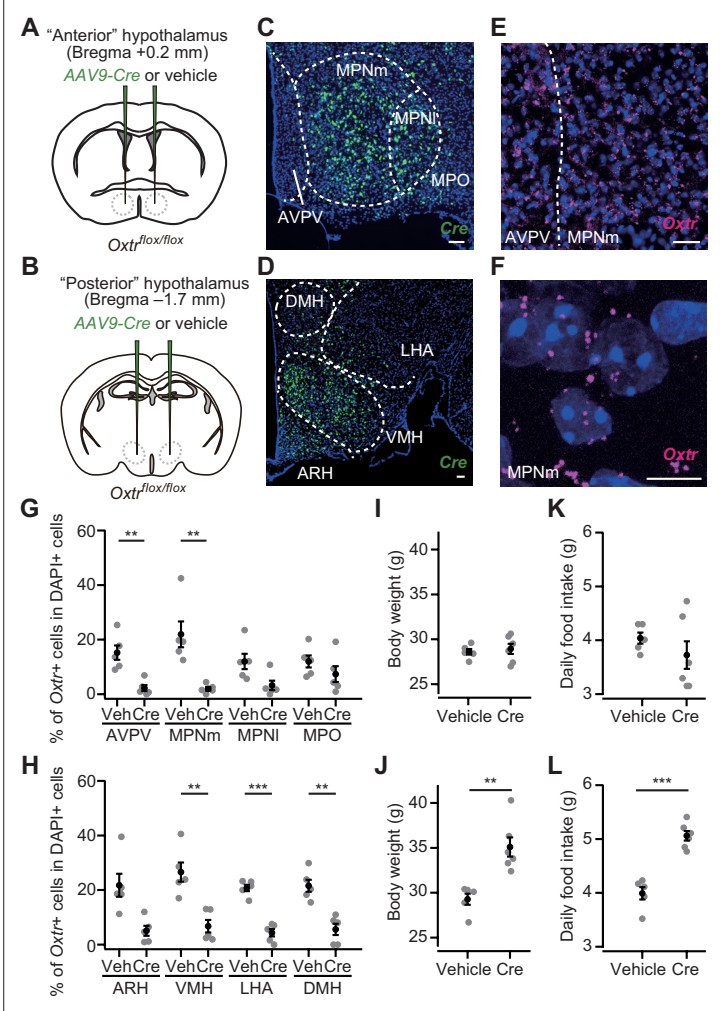

**Figure 5.** *Oxtr* conditional knockout (cKO) in the posterior hypothalamus induces increases in body weight and food intake. (**A, B**) Schematic of the virus injection. *AAV-Cre* or vehicle was injected into the bilateral anterior or posterior hypothalamus (see Materials and methods) of *Oxtr<sup>flox/flox</sup>* male mice. (**C, D**) Representative coronal section showing *Cre* mRNA (green). Blue, DAPI. Scale bar, 50 µm. (**E**) A representative coronal section showing anteroventral periventricular nucleus (AVPV) and medial preoptic nucleus medial part (MPNm) from a vehicle-injected mouse. *Oxtr* mRNA was visualized by RNAscope (magenta). Blue, DAPI. Scale bar, 30 µm. (**F**) Projection of a confocal stack in MPNm from a vehicle-injected mouse. Magenta, *Oxtr* mRNA. Blue, DAPI. Scale bar, 5 µm. (**G, H**) Fraction of DAPI+ cells expressing *Oxtr*. Cells showing three or more RNAscope dots were defined as *Oxtr*+ (*Figure 5—figure supplement 2*). Veh, vehicle. N=5 each. **$p<0.01$, ***$p<0.001$, Student's $t$-test with Bonferroni correction. Decreases in the MPN lateral part (MPNl), medial preoptic area (MPO), and arcuate hypothalamic nucleus (ARH) on average were found in *AAV-Cre*-injected mice but did not reach the level of statistical significance in Student's $t$-test with Bonferroni correction ($p=0.045$, 0.289, and 0.012, respectively). (**I, J**) Body weight measured at 5 weeks after the injection. **$p<0.01$, Student's $t$-test. Anterior hypothalamus, N=5 and 6 for vehicle and Cre, respectively, and posterior hypothalamus, N=5 and 6 for vehicle and Cre, respectively. (**K, L**) Daily food intake measured at 5 weeks after the injection. ***$p<0.001$, Student's $t$-test. Anterior hypothalamus, N=5 and 6 for vehicle and Cre, respectively, and posterior hypothalamus, N=5 and 6 for vehicle and Cre, respectively. Error bars, SEM.

The online version of this article includes the following figure supplement(s) for figure 5:

**Figure supplement 1.** Distribution of *oxytocin receptor* (*Oxtr*)+ RNAscope dots.

**Figure supplement 2.** Testing the stringency in the definition of *oxytocin receptor* (*Oxtr*)+ cell by different numbers of RNAscope dots.

### Downstream of Oxt neurons that mediate appetite suppression signals

After eating a sufficient amount of food, animals stop eating owing to the appetite suppression signals mediated in the brain. Several brain regions and cell types, such as Pomc-expressing neurons in the ARH (*Ollmann et al., 1997*; *Sternson and Eiselt, 2017*; *Sutton et al., 2016*), glutamatergic *Oxtr*-expressing neurons in the ARH (*Fenselau et al., 2017*), and calcitonin gene-related peptide expressing neurons in the parabrachial nucleus (*Carter et al., 2013*; *Wu et al., 2012a*), have been identified in this process. Our data showed that PVH Oxt neurons mediated appetite suppression signals. Previous studies have shown that both oxytocinergic neurites and *Oxtr*-expressing neurons can be found in various brain and spinal cord regions (*Jurek and Neumann, 2018*; *Lefevre et al., 2021*; *Newmaster et al., 2020*; *Oti et al., 2021*; *Zhang et al., 2021*). In the present study, by *AAV-Cre*-mediated cKO, we found that *Oxtr* expressed by neurons in the posterior hypothalamic regions, especially those in the ARH, mediates appetite suppression signals (*Figures 5 and 6*). Our data are, therefore, generally consistent with the view that *Oxtr*-expressing neurons in the ARH evoke satiety signaling (*Fenselau et al., 2017*; *Maejima et al., 2014*); however, we do not exclude the possibility that OxtR in the other parts of the posterior hypothalamus, such as the VMH (*Leng et al., 2008*; *Viskaitis et al., 2017*) and medulla (*Ong et al., 2017*), also contributes to appetite suppression. Collectively, we suggest that one of the output pathways of the PVH for body weight homeostasis is mediated by Oxt signaling-based modulation of other hypothalamic appetite regulation systems.

We also showed that ip administration of Oxt can mitigate the overeating phenotype caused by the PVH *Oxt* cKO model (*Figure 4*). Together with the importance of OxtR signaling in the ARH, one possibility is that the primary hypothalamic neurons that are located outside the blood-brain barrier (*Yulyaningsih et al., 2017*) directly receive ip-injected Oxt and transmit appetite suppression signals. Alternatively, OxtR-expressing neurons in the peripheral nervous system, such as those in the vagal sensory neurons transmitting intentional appetite suppression signals (*Bai et al., 2019*), may indirectly modify feeding. Future studies should further dissect the responsible cell types and physiological functions of OxtR signaling in the ARH. Recent advances in the real-time imaging of OxtR activities, for example, with a circularly permuted green fluorescent protein binding to OxtR (*Ino et al., 2022*; *Qian et al., 2022*), would be useful for delineating the circuit mechanism and spatiotemporal dynamics of the Oxt-mediated suppression of hyperphagic obesity, such as by pinpointing the site of Oxt release.

## Materials and methods

### Key resources table

| Reagent type (species) or resource | Designation | Source or reference | Identifiers | Additional information |
|---|---|---|---|---|
| Strain, strain background (mouse, male) | *Oxt* KO | *Inada et al., 2022* | #CDB0204E | |
| Strain, strain background (mouse, male) | *Oxt* cKO (floxed) | *Inada et al., 2022* | #CDB0116E | |
| Strain, strain background (mouse, male) | *Oxtr*<sup>flox/flox</sup> | *Takayanagi et al., 2005* | | |
| Recombinant DNA reagent | AAV9-*hSyn-Cre* | Addgene | RRID:Addgene_105555-AAV9 | |
| Recombinant DNA reagent | AAV2-*CMV-Cre-GFP* | University of North Carolina viral core | | https://www.med.unc.edu/genetherapy/vectorcore/in-stock-aav-vectors/reporter-vectors/ |
| Commercial assay or kit | RNAscope Multiplex Fluorescent Reagent Kit | Advance Cell Diagnostics | 323110 | |
| Commercial assay or kit | RNAscope Mm-OXTR | Advance Cell Diagnostics | 412171 | |
| Software, algorithm | Igor Pro | Wavemetrics | RRID: SCR_000325 | |
| Software, algorithm | ImageJ | NIH | RRID: SCR_003070 | |

## Animals

All experiments were conducted with virgin male mice. Animals were housed under a 12 hr light/12 hr dark cycle with ad libitum access to water and standard mouse pellets (MFG; Oriental Yeast, Shiga, Japan; 3.57 kcal/g). Wild-type C57BL/6J mice were purchased from Japan SLC (Hamamatsu, Japan). *Oxt* KO (Accession No. CDB0204E) and cKO (Accession No. CDB0116E) lines (listed at http://www2. clst.riken.jp/arg/mutant%20mice%20list.html) were generated and validated previously (**Inada et al., 2022**). The *Oxtr^{flox/flox}* mouse line has been described (**Takayanagi et al., 2005**). *Oxtr* KO mice were generated by injecting *Cre* mRNA into the *Oxtr^{flox/flox}* zygotes. We used only the mice that had the deletion allele without the *flox* allele by genotype PCR in the analysis. We confirmed the result of *Figure 5—figure supplement 2* in a small number of *Oxtr* KO mice that had been generated by conventional crossing from Oxtr *flox* mice. All animal procedures followed the animal care guidelines approved by the Institutional Animal Care and Use Committee of the RIKEN Kobe branch.

We chose the *Oxt flox/null* model to increase the efficiency of cKO (*Figures 1–4*). If we had used the *Oxt flox/flox* mice for cKO, because of high *Oxt* gene expression levels, a small fraction of the *flox* alleles that do not experience recombination would easily mask the phenotypes. Because *flox/null* alone (without Cre) has no phenotype (*Figure 1F*), we could justify the use of the *flox/null* model. Regarding *Oxtr* cKO (*Figures 5 and 6*), we chose the *flox/flox* model because the haploinsufficiency gene effect of *Oxtr* has been reported, at least in the context of social behaviors (**Sala et al., 2013**).

## Stereotactic viral injections

We obtained the AAV serotype 9 *hSyn-Cre* from Addgene (#105555; titer: $2.3 \times 10^{13}$ genome particles/ mL) and the AAV serotype 2 *CMV-Cre-GFP* from the University of North Carolina viral core ($7.1 \times 10^{12}$ genome particles/mL). To target the AAV or saline (vehicle) into a specific brain region, stereotactic coordinates were defined for each brain region based on the Allen Mouse Brain Atlas (**Lein et al., 2007**). Mice were anesthetized with 65 mg/kg ketamine (Daiichi Sankyo, Tokyo, Japan) and 13 mg/kg xylazine (X1251; Sigma-Aldrich) via ip injection and head-fixed to stereotactic equipment (Narishige, Tokyo, Japan). The following coordinates were used (in mm from the bregma for anteroposterior [AP], mediolateral [ML], and dorsoventral [DV]): PVH, AP –0.8, ML 0.2, DV 4.5; SO, AP –0.7, ML 1.2, DV 5.5; LHA, AP –2.0, ML 1.2, DV 5.2; ARH, AP –2.0, ML 0.2, DV 5.8. We defined the anterior and posterior hypothalamus by the following coordinates: anterior, AP +0.2, ML 0.2, DV 5.2; posterior, AP –1.7, ML 1.0, DV 5.2. The injected volume of AAV was 200 nL at a speed of 50 nL/min. After viral injection, the animal was returned to the home cage. In *Figure 4*, 100 µL of Oxt (1910, Tocris) dissolved in saline (vehicle) at 500 µM or 1 mM was ip-injected once every 3 days from the next day of *AAV-Cre* injection.

## Measurement of food intake

Food intake was measured by placing pre-weighted food pellets on the plate of a cage and reweighing them. In all the experiments, except those in *Figure 4—figure supplement 1E–H*, daily food intake was measured as follows: 200 g of food pellets were placed and food intake was measured once a week (weekly food intake). Daily food intake was calculated by dividing the weekly food intake by 7 (days) and reported with significance digits of 0.1 g. In *Figure 4—figure supplement 1E–H*, after 6 hr of fasting, 100 µL of Oxt dissolved in saline (vehicle) at 500 µM or 1 mM was ip-injected. Then, 80.0 g of food was placed and food intake was measured in units of 0.1 g after 1, 3, and 5 hr.

## Fluorescent ISH

Fluorescent ISH was performed as previously described (**Inada et al., 2022**; **Ishii et al., 2017**). In brief, mice were anesthetized with sodium pentobarbital and perfused with PBS followed by 4% PFA in PBS. The brain was post-fixed with 4% PFA overnight. Twenty µm coronal brain sections were made using a cryostat (Leica). The following primer sets were used in this study: *Cre* forward, CCAAGAAG AAGAGGAAGGTGTC; *Cre* reverse, ATCCCCAGAAATGCCAGATTAC; *Oxt* forward, AAGGTCGG TCTGGGCCGGAGA; and *Oxt* reverse, TAAGCCAAGCAGGCAGCAAGC. Fluoromount (K024; Diagnostic BioSystems) was used as a mounting medium. Brain images were acquired using an Olympus BX53 microscope equipped with a ×10 (NA 0.4) objective lens. Cells were counted manually using the ImageJ Cell Counter plugin. In *Figures 1C and 2C*, cells were counted by an experimenter who was blind to the experimental conditions.

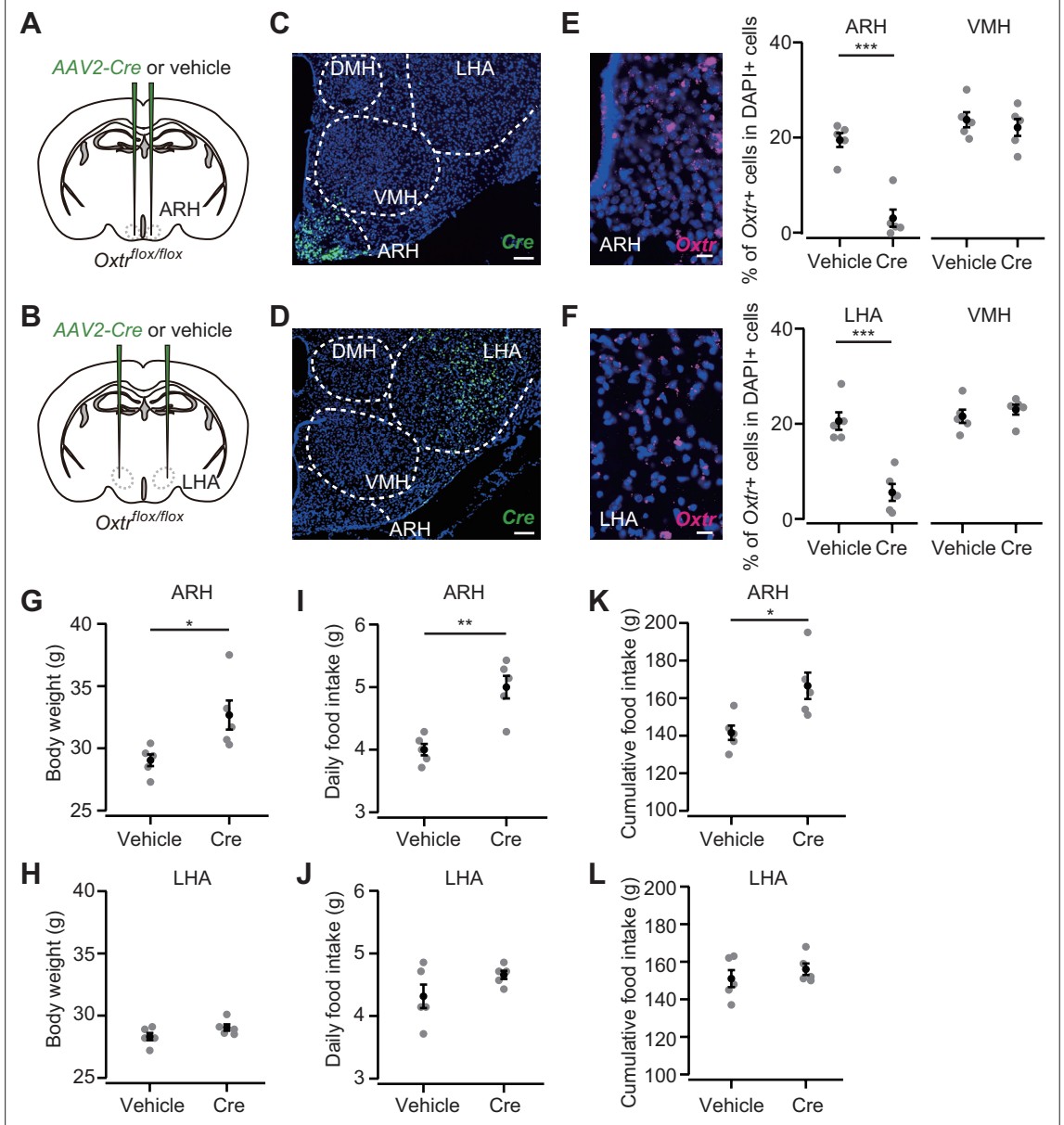

**Figure 6.** *Oxytocin receptor (Oxtr)* expression in the arcuate hypothalamic nucleus (ARH) suppresses body weight and food intake. (**A, B**) Schematic of the virus injection. *AAV-Cre* or vehicle was injected into the bilateral ARH or lateral hypothalamic area (LHA) of *Oxtr^flox/flox* male mice. (**C, D**) Representative coronal section showing *Cre* mRNA (green). Blue, DAPI. Scale bar, 50 µm. (**E, F**) Left, representative coronal section showing the ARH or LHA from a vehicle-injected mouse. *Oxtr* mRNA was visualized by RNAscope (magenta). Blue, DAPI. Scale bar, 5 µm. Right, fraction of DAPI+ cells expressing *Oxtr* in the ARH (**E**) or LHA (**F**) and ventromedial hypothalamic nucleus (VMH), a neighboring nucleus of the ARH and LHA. Cells showing three or more RNAscope dots were defined as *Oxtr+*. N=5 each. ***p<0.001, Student's *t*-test with Bonferroni correction. (**G, H**) Body weight measured at 5 weeks after the injection. *p<0.05, Student's *t*-test. N=5 each. (**I, J**) Daily food intake measured at 5 weeks after the injection. **p<0.01, Student's *t*-test. N=5 each. (**K, L**) Cumulative food intake during the 5 weeks after the injection. *p<0.05, Student's *t*-test. N=5 each. Error bars, SEM.

## RNAscope assay

*Oxtr* mRNA was visualized by the RNAscope Multiplex Fluorescent Reagent Kit (323110; Advance Cell Diagnostics [ACD]) according to the manufacturer's instructions. In brief, 20 µm coronal brain sections were made using a cryostat (Leica). A probe against *Oxtr* (Mm-OXTR, 412171, ACD) was hybridized in a HybEZ Oven (ACD) for 2 hr at 40°C. Then, the sections were treated with TSA-plus Cyanine 3 (NEL744001KT; Akoya Biosciences; 1:1500). Fluoromount (K024; Diagnostic BioSystems) was used as a mounting medium. Images subjected to the analysis were acquired using an Olympus BX53

microscope equipped with a ×10 (NA 0.40) or ×20 (NA 0.75) objective lens, as shown in *Figures 5E, 6E and F*. *Figure 5F* was obtained with a confocal microscope (LSM780, Zeiss) equipped with a ×63 oil-immersion objective lens (NA 1.40). RNAscope dots were counted manually using the ImageJ Cell Counter plugin.

## Measurements of the weight of livers and stomachs

Mice were anesthetized with sodium pentobarbital. Livers and stomachs were obtained without perfusion and their weight was immediately measured. Before measurement, the stomach was gently pressed to eject the remaining contents.

## Plasma measurements

Mice were fed ad libitum before blood sampling. Mice were anesthetized with isoflurane and blood was collected from the heart. EDTA was used to prevent blood coagulation. Plasma concentrations of glucose, triglycerides, and leptin were measured by the enzymatic method, HK-G6PDH, and ELISA, respectively, through a service provided by Oriental Yeast (Shiga, Japan).

## Data analysis

All mean values are reported as the mean ± SEM. The statistical details of each experiment, including the statistical tests used, the exact value of n, and what n represents, are shown in each figure legend. The p-values are shown in each figure legend or panel; nonsignificant values are not noted. In *Figures 1G and 2E*, exponential fit was calculated by Igor (WaveMetrics).

## Acknowledgements

We wish to thank Mitsue Hagihara for the extensive technical support, the Laboratory for Comprehensive Bioimaging for the microscopy services, and the members of the Miyamichi lab for their comments on an earlier version of this manuscript. This work was supported by the RIKEN Special Postdoctoral Researchers Program, a grant from the Kao Foundation for Arts and Sciences, and JSPS KAKENHI (19J00403 and 19K16303) to KI, and the JST CREST program (JPMJCR2021) and JSPS KAKENHI (20K20589) to KM.

## Additional information

### Funding

| Funder | Grant reference number | Author |
| --- | --- | --- |
| Japan Society for the Promotion of Science | KAKENHI 19J00403 | Kengo Inada |
| Japan Society for the Promotion of Science | KAKENHI 19K16303 | Kengo Inada |
| Japan Science and Technology Agency | CREST JPMJCR2021 | Kazunari Miyamichi |
| Japan Society for the Promotion of Science | KAKENHI 20K20589 | Kazunari Miyamichi |

The funders had no role in study design, data collection and interpretation, or the decision to submit the work for publication.

### Author contributions

Kengo Inada, Conceptualization, Formal analysis, Funding acquisition, Investigation, Methodology, Writing – original draft, Project administration, Writing – review and editing; Kazoku Tsujimoto, Formal analysis; Masahide Yoshida, Katsuhiko Nishimori, Resources; Kazunari Miyamichi, Conceptualization, Supervision, Funding acquisition, Writing – original draft, Project administration, Writing – review and editing

Author ORCIDs
Kengo Inada (iD) http://orcid.org/0000-0002-0859-4582
Masahide Yoshida (iD) http://orcid.org/0000-0003-1247-2294
Kazunari Miyamichi (iD) http://orcid.org/0000-0002-7807-8436

Ethics
All animal procedures followed animal care guidelines approved by the Institutional Animal Care and Use Committee of the RIKEN Kobe branch (#A2017-15-12).

Decision letter and Author response
Decision letter https://doi.org/10.7554/eLife.75718.sa1
Author response https://doi.org/10.7554/eLife.75718.sa2

## Additional files

Supplementary files
• Transparent reporting form

Data availability
All data generated or analyzed during this study are included in the manuscript and the figure supplement.

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
