## [Editor Report]

Inada and colleagues report results from a series of studies investigating the role of the neuropeptide oxytocin in regulating food intake and body weight. They used combinations of genetics and behavioral studies to demonstrate that oxytocin deletion results in increased food intake and body weight. They further show that deletion of the oxytocin receptor in the posterior hypothalamus causes a similar increase in food intake and body weight. Collectively, these studies support a role for the oxytocin system as a key regulator of energy balance.

---

## [Decision Letter]

**Decision letter after peer review:**

Thank you for submitting your article "Oxytocin signaling in the posterior hypothalamus prevents hyperphagic obesity in mice" for consideration by *eLife*. Your article has been reviewed by 3 peer reviewers, one of whom is a member of our Board of Reviewing Editors, and the evaluation has been overseen by Ma-Li Wong as the Senior Editor. The reviewers have opted to remain anonymous.

Essential revisions:

The reviewers agree that several technical issues outlined below need to be addressed. In particular, the authors should address the sample sizes noted below.

In addition, more precise description of the anatomic details is needed throughout the manuscript.

*Reviewer #1 (Recommendations for the authors):*

(1) It is not clear the reason behind using the partial conditional KO mice, OTflox/-. Could you explain why flox/KO was used instead of flox/flox model? If so, these mice are "partial" conditional KO model instead of "bona fida" conditional KO, aren't they? The authors need to clarify this model.

(2) Regarding Figure 4, previous literature has shown that OT injection reduces food intake within an hour. Does OT injection into OT knockout in the PVH mice have the same effect? It is important as these type of experiments can point out that OT may be required for transcriptional changes and/or circuit re-wiring (and others) to regulate food intake.

(3) A large body of studies have shown that the ventrolateral side of the ventromedial hypothalamic nucleus (VMHvl) regulates food intake. In addition, recent optogenetic and chemogenetic studies have shown that the dorsomedial and center part of VMH (VMHdm/c) also can regulate food intake, although many genetic studies (e.g., knockout the important genes such as leptin receptors from the VMHdm/c) do not support this notion. The authors pinpointed that the phenotype in Figure 5 resulted from the ablation of OTR in the arcuate hypothalamic nucleus (ARH) and/or the lateral hypothalamic area (LHA). However, Cre-GFP expression was strongly observed throughout the VMH (Figure 5D). The authors may want to discuss/address the possibility that OTR in the VMH may be involved in OT-induced food suppression effects.

*Reviewer #2 (Recommendations for the authors):*

This is a straightforward and important study investigating the role of oxytocin signaling on appetite and body weight. By loss of function in adult mice, the authors show a clear role for oxytocin as a inhibitory molecule for overeating.

The major critiques of the study are

1. The sample sizes are typically too small (often n=4), especially in Figure 4 where the effect sizes are smaller.

2. The authors have not ruled out the possibility that viral overexpression of Cre in the PVH is responsible for the hyperphagia. Testing this in the OT-/- and in OT+/+ mice would be prudent. There is no information about the viral titer or surgical procedures, but I suspect high viral titers were sued, which could non-specifically reduced PVH function.

3. The fraction of neurons reported to be OXTR+ seems too high. The authors should determine the background rate of OXTR detection in OXTR-/- mice or in regions known not to expression OXTR and use this a threshold for OXTR expression vs noise e.g., lipofuscin can show up as a false positive in RNAscope and this is determined by measuring if the spot shows broadband fluorescence at other wavelengths.

*Reviewer #3 (Recommendations for the authors):*

The manuscript by Inada et al. examines the role of hypothalamic oxytocin (OT) signaling in feeding behavior. They demonstrate that conditional knockout of OT in the adult paraventricular hypothalamic nucleus (PVH) increases body weight through increases in food intake, and that conditional knockout of the OT receptor in the posterior hypothalamus has a similar effect. The authors therefore conclude that OT signaling in the posterior hypothalamus, presumably through oxytocin produced in the PVH, contributes to energy balance control.

Strengths:

There has been conflicting literature on the role of OT in feeding behavior. Although pharmacological and genetic approaches have suggested an anorexic effect of OT, knockout of OT or OT receptor has minimal effect on feeding. To address this apparent discrepancy, the authors use conditional knockout models to manipulate OT signaling. This allows not only temporal control of OT and OT receptor, but also allows investigation of signaling in different brain regions (versus, for example, whole body or organ). That the conditional knockout mice display hyperphagia and obesity begins to settle this conflict in the literature.

Weaknesses:

1) There is not a major conceptual advance. The data largely confirm what pharmacological and RNAi knockdown studies have previously demonstrated.

2) The finding that IP injection of OT partially rescues the phenotype of the KO mouse lacks rigor and proper controls.

• Only one dose of OT was used.

• The effect of IP OT in WT mice should be shown. If the IP OT reduces food intake in the WT mouse, then it could be an independent effect.

• Injection of OT within regions of the hypothalamus would provide more information about the localization of these effects.

• How do the authors reconcile the fact that there is no effect on feeding behavior but an effect on body weight, especially when the OT KO has a feeding phenotype?

3) There is little anatomical precision in the manipulation of OT receptors in the posterior hypothalamus." Understanding which of these brain regions (e.g. ARH, VMH, LHA, DMH, others?) is involved in mediating these effects would be very helpful. It is also a missed opportunity to not explore hindbrain OT receptor, as this would be a good positive control (previous literature).

4) Attempts to connect the OT KO data with the OT receptor KO data would greatly enhance this manuscript by delineating a circuit mechanism for central OT-induced anorexia. This could be achieved by giving central injections of OT into target regions of OT KO mice, for example.

Other comments:

1) Unless I am mistaken, the authors cite a biorxiv paper for the "previously-validated" mouse, but this has not been peer reviewed.

2) What dose of OT was used in Figure 4? Please provide in g.

3) How was food intake measured?

4) The authors say in the text that the changes in the weight of the viscera and the blood constituents were independent of body weight. However, the legend says that these measurements were taken 5 weeks post injection, and Figure 1 shows a clear BW phenotype at 5 weeks post-injection. Please clarify.

---

## [Author Response]

Essential revisions:The reviewers agree that several technical issues outlined below need to be addressed. In particular, the authors should address the sample sizes noted below.In addition, more precise description of the anatomic details is needed throughout the manuscript.Reviewer #1 (Recommendations for the authors):(1) It is not clear the reason behind using the partial conditional KO mice, OTflox/-. Could you explain why flox/KO was used instead of flox/flox model? If so, these mice are "partial" conditional KO model instead of "bona fida" conditional KO, aren't they? The authors need to clarify this model.

We thank the reviewer for the positive evaluation of our main aims and conclusions, and for very constructive suggestions. We chose the Oxt flox/null model to increase the efficiency of Cre-mediated KO. If we had used Oxt flox/flox mice, which have very high Oxt gene expression levels, a small fraction of the flox alleles that do not experience recombination would easily mask the body weight and feeding phenotypes. Because flox/null alone (without Cre) has no phenotype (Figure 1F), we could justify the use of the flox/null model. In principle, this situation is comparable to analyzing homozygous mutants by using heterozygotes as a reference, which is common in the genetics of mice and flies. In addition, many influential papers have utilized flox/null models as conditional knockout (for example, Kwan et al. Genesis 39, 10, 2004; Pelosi et al., Plos One 10: e0136422, 2015). In the revised manuscript, we added additional explanations in the Materials and methods to clarify our thoughts (lines 312–318). Regarding Oxtr conditional knockout, we chose the flox/flox model because the haploinsufficiency gene effect of Oxtr has been reported, at least in the context of social behaviors (Sala et al. J Neuroendocrinology 25, 107, 2013).

(2) Regarding Figure 4, previous literature has shown that OT injection reduces food intake within an hour. Does OT injection into OT knockout in the PVH mice have the same effect? It is important as these type of experiments can point out that OT may be required for transcriptional changes and/or circuit re-wiring (and others) to regulate food intake.

We deeply appreciate this important comment. According to this suggestion, we conducted a new experiment to test the short-term effect of IP injection of Oxt. We found that Oxt cKO mice ate significantly less after 1, 3, and 5 hours of Oxt IP injection, consistent with previous literature using wild-type rats or mice. We reported these data in new Figure 4—figure supplement 1E–H.

(3) A large body of studies have shown that the ventrolateral side of the ventromedial hypothalamic nucleus (VMHvl) regulates food intake. In addition, recent optogenetic and chemogenetic studies have shown that the dorsomedial and center part of VMH (VMHdm/c) also can regulate food intake, although many genetic studies (e.g., knockout the important genes such as leptin receptors from the VMHdm/c) do not support this notion. The authors pinpointed that the phenotype in Figure 5 resulted from the ablation of OTR in the arcuate hypothalamic nucleus (ARH) and/or the lateral hypothalamic area (LHA). However, Cre-GFP expression was strongly observed throughout the VMH (Figure 5D). The authors may want to discuss/address the possibility that OTR in the VMH may be involved in OT-induced food suppression effects.

We are greatly thankful for this important comment. We have conducted additional experiments to remove the Oxtr gene with a high spatial resolution (now reported in new Figure 6). To achieve this, we injected serotype 2 of AAV-Cre into the ARH or LHA. As expected, Oxtr expression in the targeted nucleus was significantly reduced, while that in the VMH, a neighboring nucleus, was unaffected (Figure 6E and 6F). We also found an increase in food intake and body weight after injecting the AAV into the ARH, but not into the LHA. Still, these results do not rule out the possibility that VMH contributes to the suppression of hyperphagic obesity. Nevertheless, we appreciate the importance of VMH for feeding regulation and added a discussion regarding VMH in the revised manuscript (lines 278–280).

Reviewer #2 (Recommendations for the authors):This is a straightforward and important study investigating the role of oxytocin signaling on appetite and body weight. By loss of function in adult mice, the authors show a clear role for oxytocin as a inhibitory molecule for overeating.The critiques of the study are1. The sample sizes are typically too small (often n=4), especially in Figure 4 where the effect sizes are smaller.

We are greatly thankful to this reviewer for the positive evaluation of our main aims and conclusions, as well as for the very constructive suggestions. As described in the revised manuscript, we have included additional cohorts of Oxt cKO (Figure 1) and IP injection of Oxt (Figure 4) to collect enough animals. In addition, in Figure 4, we added different doses of Oxt IP injection, which further strengthens our initial conclusion.

2. The authors have not ruled out the possibility that viral overexpression of Cre in the PVH is responsible for the hyperphagia. Testing this in the OT-/- and in OT+/+ mice would be prudent. There is no information about the viral titer or surgical procedures, but I suspect high viral titers were sued, which could non-specifically reduced PVH function.

We appreciate this important comment. According to this suggestion, we injected 200 nL of *AAV* serotype *9 hSyn-Cre* (2.3 x 10^13^ genome particles/mL) into wild-type PVH. Neither body weight nor daily food intake measured at 5 weeks after the injection differed from the saline-injected control. This result is reported in new Figure 1—figure supplement 1, and the experimental conditions are described in the Materials and methods.

3. The fraction of neurons reported to be OXTR+ seems too high. The authors should determine the background rate of OXTR detection in OXTR-/- mice or in regions known not to expression OXTR and use this a threshold for OXTR expression vs noise e.g., lipofuscin can show up as a false positive in RNAscope and this is determined by measuring if the spot shows broadband fluorescence at other wavelengths.

We sincerely thank the reviewer for this important comment. We newly generated *Oxtr^–/–^* mice by injecting Cre mRNA into Oxtr flox/flox zygotes to conduct negative control experiments of RNAscope assay. In agreement with the concern raised by the reviewer, 1 or 2 RNAscope dots were detected in approximately 25% of DAPI+ cells of *Oxtr^–/–^* mice, as reported in new Figure 5—figure supplement 1. This means that Figure 5 of our initial manuscript contained a substantial fraction of pseudo-positive detection of Oxtr. Because we found that defining Oxtr+ cells as a DAPI+ cell with three or more dots provides a near-zero ratio (less than 5%) of pseudo-positive detection of Oxtr+ cells in *Oxtr^–/–^* mice (Figure 5—figure supplement 2), we applied this new definition throughout the paper (Figures 5 and 6). Regardless of this revised definition, our conclusion remains the same, given that AAV-Cre injection into Oxtr^flox/flox^ mice significantly reduced the number of Oxtr+ cells (Figures 5 and 6).

Reviewer #3 (Recommendations for the authors):The manuscript by Inada et al. examines the role of hypothalamic oxytocin (OT) signaling in feeding behavior. They demonstrate that conditional knockout of OT in the adult paraventricular hypothalamic nucleus (PVH) increases body weight through increases in food intake, and that conditional knockout of the OT receptor in the posterior hypothalamus has a similar effect. The authors therefore conclude that OT signaling in the posterior hypothalamus, presumably through oxytocin produced in the PVH, contributes to energy balance control.Strengths:There has been conflicting literature on the role of OT in feeding behavior. Although pharmacological and genetic approaches have suggested an anorexic effect of OT, knockout of OT or OT receptor has minimal effect on feeding. To address this apparent discrepancy, the authors use conditional knockout models to manipulate OT signaling. This allows not only temporal control of OT and OT receptor, but also allows investigation of signaling in different brain regions (versus, for example, whole body or organ). That the conditional knockout mice display hyperphagia and obesity begins to settle this conflict in the literature.Weaknesses:1) There is not a major conceptual advance. The data largely confirm what pharmacological and RNAi knockdown studies have previously demonstrated.

We are sincerely thankful to this reviewer for the positive evaluation of our main aims and conclusions, as well as for the very constructive suggestions. We agree with the view that the main aim of this study is to resolve the conflict between genetic mutant and pharmacological/knockdown studies. However, in the revised manuscript, we have provided additional evidence that OxtR signaling, specifically in the arcuate nucleus, is responsible for suppressing hyperphagia (as reported in new Figure 6). We think that the demonstration of such functional localization of Oxt-OxtR signaling can provide some biological advances, as it cannot be achieved easily by classical pharmacology.

2) The finding that IP injection of OT partially rescues the phenotype of the KO mouse lacks rigor and proper controls.• Only one dose of OT was used.• The effect of IP OT in WT mice should be shown. If the IP OT reduces food intake in the WT mouse, then it could be an independent effect.• Injection of OT within regions of the hypothalamus would provide more information about the localization of these effects.• How do the authors reconcile the fact that there is no effect on feeding behavior but an effect on body weight, especially when the OT KO has a feeding phenotype?

We are also thankful for this important comment. According to these suggestions, we have included an additional cohort of IP injection of Oxt into Oxt cKO mice to test different doses of Oxt, as reported in Figure 4 in the revised manuscript. Both 50.36 µg and 100.72 µg per injection significantly reduced body weight and food intake in Oxt cKO mice. Importantly, IP injection of the same dose of Oxt into wild-type mice did not affect body weight or food intake, as reported in new Figure 4—figure supplement 1A–D. With the greater number of animals in multiple conditions, the reduced feeding by IP injection of Oxt is more explicitly shown in the revised manuscript. Still, the plasma triglyceride and leptin concentrations were not fully rescued by IP injection of Oxt in Oxt cKO mice, suggesting that the Oxt provided via IP injection is somehow suboptimal in regard to its effect site in the body and/or brain.

Regarding the functional identification of the effective site of Oxt, we decided to conduct local cKO of Oxtr (see response to Reviewer3, point 3 below) instead of local infusion of Oxt into the target brain regions (such as ARH). This is because (i) it is generally difficult to evaluate the diffusion of a solution containing Oxt after local infusion, and (ii) animals need to be anesthetized repeatedly for local infusion of Oxt through a guide cannula, which by itself could affect their feeding patterns.

3) There is little anatomical precision in the manipulation of OT receptors in the posterior hypothalamus." Understanding which of these brain regions (e.g. ARH, VMH, LHA, DMH, others?) is involved in mediating these effects would be very helpful. It is also a missed opportunity to not explore hindbrain OT receptor, as this would be a good positive control (previous literature).

We appreciate this comment. Although precisely controlling Cre activity at the resolution of a single anatomical structure in the brain is not trivial, by using AAV serotype 2, which tends to spread less around the injection site, we conducted ARH- and LHA-specific loss-of-function of Oxtr (new Figure 6). We found that body weight, daily food intake, and cumulative food intake were significantly larger in the mice that received AAV-Cre injection into the ARH, whereas no significant difference was found in the mice that expressed Cre into the LHA. Although we do not exclude the contribution of Oxtr in other brain regions, these data suggest that the majority of effects given rise to by Oxtr cKO in the posterior hypothalamus can be explained by the reduction of Oxtr expression in the ARH.

4) Attempts to connect the OT KO data with the OT receptor KO data would greatly enhance this manuscript by delineating a circuit mechanism for central OT-induced anorexia. This could be achieved by giving central injections of OT into target regions of OT KO mice, for example.

We agree with this view that a circuit mechanism by which PVH Oxt neurons are linked to the posterior hypothalamus Oxtr-expressing neurons would strengthen the manuscript. However, it is highly challenging to pinpoint the site of Oxt release for preventing hyperphagia. Although our Oxtr cKO in the ARH (Figure 6) improved the spatial resolution of the site of Oxt action, this Oxtr signaling may be mediated by diffusion of Oxt from long range, such as cell bodies/dendrites in the PVH or Oxt-positive fibers elsewhere in the brain, rather than direct axonal projection from PVH Oxt neurons to the ARH. As IP-injected Oxt is also functional (Figure 4), in principle, it is possible that circulating Oxt in the bloodstream is responsible for the Oxt signaling in the ARH, presumably via a leaky blood–brain barrier or the tanycytic-mediated active transport of humoral signals. Future studies, including real-time imaging of Oxt ligand release by an Oxt sensor, would further reveal circuit mechanisms. In the revised manuscript, we have improved the spatial resolution of our Oxtr cKO (see our response #3-3). We have also mentioned that the suggested point is an important topic for future research in the Discussion (lines 293–295).

Other comments:1) Unless I am mistaken, the authors cite a biorxiv paper for the "previously-validated" mouse, but this has not been peer reviewed.

We apologize that the citation was a preprint at the time of initial submission. During the revision, it was updated as follows.

Inada K et al., Neuron 110, 2009-2023.e5, 2022. https://pubmed.ncbi.nlm.nih.gov/35443152/.

We have also updated this citation in the revised manuscript.

2) What dose of OT was used in Figure 4? Please provide in g.

According to this comment, we added a description in Figure 4.

3) How was food intake measured?

We apologize that our description in the Materials and methods was not sufficient. In the revised manuscript, we have provided further details of the food intake measurements (lines 337–345).

4) The authors say in the text that the changes in the weight of the viscera and the blood constituents were independent of body weight. However, the legend says that these measurements were taken 5 weeks post injection, and Figure 1 shows a clear BW phenotype at 5 weeks post-injection. Please clarify.

We do not think that viscera and blood constituents are independent of body weight. We apologize that our initial description was unclear. We have clarified this in the revised manuscript (lines 155–158).